## SHORT REPORT

# An *A. thaliana* mutant lacking all nine ATG8 isoforms provides genetic evidence for functional specialization of ATG8 in plants

Alessia Del Chiaro[1,2,*], Nenad Grujic[1,*], Jierui Zhao[1,2], Ranjith Kumar Papareddy[1,3], Peng Gao[1], Juncai Ma[4], Christian Lofke[1], Anuradha Bhattacharya[1], Ramona Gruetzner[5], Pierre Bourguet[1], Frédéric Berger[1], Byung-Ho Kang[4], Sylvestre Marillonnet[5] and Yasin Dagdas[1,3,‡]

## ABSTRACT

Autophagy sustains cellular health by recycling damaged or excess components through autophagosomes. Autophagy is mediated by conserved ATG proteins, among which the ubiquitin-like ATG8 proteins play a central role by linking cargo to the growing autophagosomes. Unlike most ATG proteins, the ATG8 gene family is significantly expanded in vascular plants, but its functional specialization remains poorly understood. Using transcriptional and translational reporters in *Arabidopsis thaliana*, we revealed that ATG8 isoforms are differentially expressed across tissues and form distinct autophagosomes. To explore ATG8 specialization, we generated the nonuple *Δatg8* mutant, lacking all nine ATG8 isoforms. The mutant displayed hypersensitivity to carbon and nitrogen starvation, coupled with defects in bulk and selective autophagy, as shown by biochemical and ultrastructural analyses. Complementation experiments demonstrated that ATG8A could rescue both carbon and nitrogen starvation phenotypes, whereas ATG8H could only complement carbon starvation. Proximity labeling proteomics further identified isoform-specific interactors under nitrogen starvation, underscoring their functional divergence. These findings provide genetic evidence for functional specialization of ATG8 isoforms in plants and lay the foundation for investigating their roles in diverse cell types and stress conditions.

KEY WORDS: Autophagy, *Arabidopsis*, Atg8, Selective autophagy

## INTRODUCTION

Autophagy is an evolutionarily conserved cellular quality control mechanism essential for maintaining homeostasis and adapting to environmental stresses (Bassham et al., 2006; Liu and Bassham, 2012; Su et al., 2020). It functions by selectively degrading and recycling damaged, redundant or harmful cellular components, ensuring cellular integrity and energy balance (Zhou et al., 2013; Izumi et al., 2020; Marshall and Vierstra, 2018). Although autophagy occurs constitutively at basal levels, it is highly inducible under stress conditions, such as upon nutrient deprivation, hypoxia or infection (Merkulova et al., 2014; Chen et al., 2015; Ismayil et al., 2020; Wang et al., 2018). During these challenges, autophagy sustains survival by facilitating the degradation of intracellular material, which is sequestered into specialized double-membrane compartments called autophagosomes (Le Bars et al., 2014; van Doorn and Papini, 2013). These structures subsequently fuse with lytic organelles – the vacuole in plants and yeast or the lysosome in animals – where their contents are broken down and recycled (Liu and Bassham, 2012).

Contrary to initial views of autophagy as a non-selective bulk degradation process, it is now recognized as a highly selective pathway (Jin et al., 2013; Stolz et al., 2014; Farré and Subramani, 2016; Lamark and Johansen, 2021). This selectivity is mediated by specific interactions between cargo receptors, known as selective autophagy receptors (SARs), and autophagy-related proteins, such as ATG8 proteins (Johansen and Lamark, 2011; Rogov et al., 2014). These interactions enable the precise targeting of a wide range of substrates, from protein aggregates to damaged organelles, thus tailoring autophagic responses to specific cellular needs (Luo et al., 2021).

Autophagosome biogenesis progresses through three tightly regulated stages: initiation, expansion and maturation. This process is orchestrated by the autophagy-related (ATG) protein family, comprising ∼40 conserved members (Wen and Klionsky, 2016; Yamamoto et al., 2023). Central to autophagy are ATG8 proteins, ubiquitin-like proteins crucial for autophagosome formation, cargo recruitment and membrane trafficking (Kaufmann and Wollert, 2014; Kaufmann et al., 2014). Once processed and lipidated, ATG8 associates with the autophagosome membrane, acting as a scaffold for the assembly of other core autophagy machinery and cargo receptors (Johansen and Lamark, 2020; Liu et al., 2021).

Interestingly, unlike most ATG proteins, which exist as single or a few isoforms, the ATG8 gene family has undergone significant expansion in vascular plants (Popelka and Klionsky, 2015; Seo et al., 2016). Whereas yeast and eukaryotes from early branching groups encode a single ATG8, vascular plants possess multiple isoforms, with *Arabidopsis thaliana* containing nine distinct ATG8 genes (AtATG8A–ATG8I) forming two major clades. Clade I contains ATG8A–ATG8G, whereas Clade II contains ATG8H and ATG8I (Kellner et al., 2017). Despite moderate sequence divergence and differential expression patterns (Winter et al., 2007; Sláviková et al., 2005; Thompson et al., 2005), the biological significance of ATG8 isoform expansion and its implications for selective autophagy remain poorly understood.

Here, we present genetic evidence for functional specialization of ATG8 isoforms in *A. thaliana* by generating a nonuple ATG8

[1]Gregor Mendel Institute, Austrian Academy of Sciences, Vienna BioCenter, Vienna 1030, Austria. [2]Vienna BioCenter PhD Program, Doctoral School of the University at Vienna and Medical University of Vienna, Vienna 1030, Austria. [3]Heidelberg University, Centre for Organismal Studies (COS), 69120 Heidelberg, Germany. [4]School of Life Sciences, Centre for Cell & Developmental Biology and State Key Laboratory of Agrobiotechnology, The Chinese University of Hong Kong, Shatin, New Territories, Hong Kong, China. [5]Leibniz-Institut für Pflanzenbiochemie, Department of Cell and Metabolic Biology, Halle 06120, Germany.
*These authors contributed equally to this work

‡Author for correspondence (yasin.dagdas@cos.uni-heidelberg.de)

A.D., 0009-0002-1341-5195; N.G., 0009-0002-1634-1507; J.Z., 0000-0002-6690-4040; R.K.P., 0000-0002-4067-5999; P.G., 0000-0001-6633-0967; P.B., 0000-0002-5019-4179; B.-H.K., 0000-0002-4299-2170; S.M., 0000-0002-4263-797X; Y.D., 0000-0002-9502-355X

mutant lacking all nine ATG8-encoding genes. Our study reveals that ATG8 isoforms not only exhibit distinct tissue-specific expression and subcellular localization but also differ in their ability to mediate autophagic responses under specific stress conditions. These findings highlight the complex regulatory landscape of autophagy in plants and provide a foundation for uncovering the mechanisms underlying ATG8 specialization.

## RESULTS

### *A. thaliana* ATG8 isoforms exhibit tissue-specific expression patterns and form distinct autophagosomes within root cells

To explore ATG8 specialization, we first checked the expression patterns of all the nine *Arabidopsis* ATG8 isoforms. We generated ATG8 promoter-GFP-GUS (pATG8X::GFP-GUS)-expressing lines and performed β-glucuronidase (GUS) staining (Fig. 1A). Interestingly, AtATG8s exhibited distinct expression patterns, and we only observed a partial overlap between different isoforms, indicating a certain degree of tissue specificity. ATG8E, ATG8F and ATG8G show a more widespread expression pattern over different tissues and organs, whereas other isoforms, including ATG8A, ATG8B and ATG8I, appear to be restricted to the root (Fig. 1A). Furthermore, some isoforms exhibit peculiar tissue- or cell-type specificities, such as ATG8D being strongly induced in the apex of the cotyledon or ATG8C appearing specifically expressed in guard cells (Fig. 1A). These observations hint at a potential tissue- or cell-type-specific function of different ATG8 isoforms.

These results are in agreement with the DNA microarray analysis of AtATG8 expression levels in different tissues reported by Thompson et al. (2005), who indicated ATG8E and ATG8F isoforms as being most highly expressed in the root. However, it should be noted that the transcript abundance might not reflect the actual protein levels, as the ATG8 isoforms could be subjected to different rates of protein turnover. In this regard, Boyecheva Woltering and Isono (2020) showed that the mRNA and protein abundance of different ATG8s in various *Arabidopsis* tissues are often not correlated. In addition, Chung et al. (2008) reported that transcript levels of ATG8 isoforms in maize respond differently to nitrogen or carbon starvation, reinforcing the idea that different ATG8 isoforms might have specialized functions.

Next, we decided to test the subcellular compartmentalization of ATG8 isoforms. We co-expressed an mCherry–ATG8E translational fusion construct with GFP-tagged versions of ATG8A, ATG8D and ATG8I, and assessed their co-localization upon treatment with the bulk autophagy-inducing chemical Torin 1 (Thoreen et al., 2020) or the ER stress inducer tunicamycin (Stephani et al., 2020) (Fig. 1B,C). Irrespective of the treatment, ATG8E colocalized almost entirely with ATG8A. Conversely, ATG8D and ATG8I exhibited a lower degree of colocalization with ATG8E after Torin 1 treatment, with an even greater reduction observed following tunicamycin treatment. ATG8I, representative of the clade II of ATG8 isoforms, has the weakest level of colocalization with ATG8E. These data indicate there are distinct pools of autophagosomes that are labeled with different ATG8 isoforms. In sum, the expression site polymorphism, as well as having varying levels of colocalization upon different treatments, support the hypothesis that ATG8 isoforms might fulfill different functions or respond to different stimuli.

### The *atg8* nonuple mutant is deficient in autophagy

The partial overlap in expression and colocalization patterns of ATG8 isoforms prompted us to generate an ATG8-free *Arabidopsis thaliana* line that we could use as a tool to investigate ATG8 specialization. Using multiplex CRISPR mutagenesis, we combined nine guide RNAs (one for each ATG8 isoform) in a construct containing an

intronized Cas9 (Grützner et al., 2021) and transformed it to generate a nonuple knockout (*Δatg8*) of all *atg8* genes. We first confirmed the mutations using whole-genome sequencing (Fig. 2A, Table S1), and we ensured the absence of ATG8 proteins by western blot analysis (Fig. S1A). To functionally verify *Δatg8* as an autophagy-deficient mutant, we performed the typical nutrient starvation assays (Chung et al., 2010). We transferred 9-day-old *Arabidopsis* seedlings to carbon or nitrogen deprived ½ MS liquid medium for 4 or 6 days, respectively. Similar to what is seen with the autophagy-deficient mutants *atg2* and *atg5* (Wang et al., 2011; Thompson et al., 2005), *Δatg8* plants exhibited reduced growth and discoloration of the cotyledons (Fig. 2B,C). To biochemically validate the *Δatg8* mutant, we performed autophagic flux assays under carbon and nitrogen starvation conditions, by measuring the endogenous levels of the stereotypical autophagy substrate NBR1 (Fig. 2D–G) (Svenning et al., 2011). NBR1 accumulated at a comparable level in both *Δatg8* and *atg5* under all conditions and was insensitive to concanamycin A treatment, which blocks vacuolar degradation (Krebs et al., 2010), denoting that both mutants are defective in autophagic degradation (Fig. 2D–G). Collectively, these results suggest that the *Δatg8* nonuple mutant is deficient in autophagy.

### The *Δatg8* mutant is defective in selective autophagy

Nitrogen and carbon starvation are considered to trigger bulk autophagy (Doelling et al., 2002; Xiong et al., 2005). To test whether *Δatg8* is also defective in selective autophagy, we tested its ability to perform mitophagy and pexophagy. For mitophagy, we measured the levels of the outer mitochondrial membrane voltage dependent anion channel I (VDAC) and the matrix protein isocitrate dehydrogenase (IDH) upon 2,4-dinitrophenol (DNP) treatment. DNP is an uncoupler that leads to mitochondrial depolarization and triggers mitophagy (Ma et al., 2021; Georgakopoulos et al., 2017) (Fig. 3A–C; Fig. S3A). Although both VDAC and IDH levels were decreased in the DNP-treated wild-type Col-0 plants, *Δatg8* plants behaved similarly to *atg5* and showed no change in VDAC or IDH levels (Ma et al., 2021). Likewise, when we assessed peroxisome degradation using the catalase antibody, the *Δatg8* mutant behaved similarly to the *atg5* mutant and was unable to perform pexophagy (Yoshimoto et al., 2014) (Fig. 3C,D; Fig. S3B). Although further characterization with multiple pexophagy and mitophagy inducers is required, these results suggest *Δatg8* mutant is defective in mitophagy and pexophagy.

Next, we performed transmission electron microscopy (TEM) analysis of mitophagy in *Arabidopsis* root cells. Although we could detect selective autophagosomal engulfment of mitochondria in wild-type Col-0 plants upon mitophagy induction, we did not observe mitophagosomes in *Δatg8* mutant (Fig. 3F). However, we could still detect double-membraned structures that resembled autophagosomes in the *Δatg8* mutant. These vesicles appear to non-specifically engulf various types of cellular components. Further studies are necessary to understand the nature of these compartments, but a plausible source could be provacuoles, compartments that appear during vacuole biogenesis (Kamada et al., 2010). Indeed, provacuole formation has been reported to be independent of autophagy, as autophagy-deficient mutants can still form provacuoles (Viotti et al., 2013). In summary, our data suggest that the *Δatg8* mutant is unable to carry out autophagic recycling.

### Complementation of *Δatg8* with ATG8A or ATG8H reveals functional specialization

After confirming the autophagy-deficient phenotype of the *Δatg8* mutant, to assess the functional specialization of ATG8 isoforms, we complemented it with GFP tagged ATG8A and ATG8H,

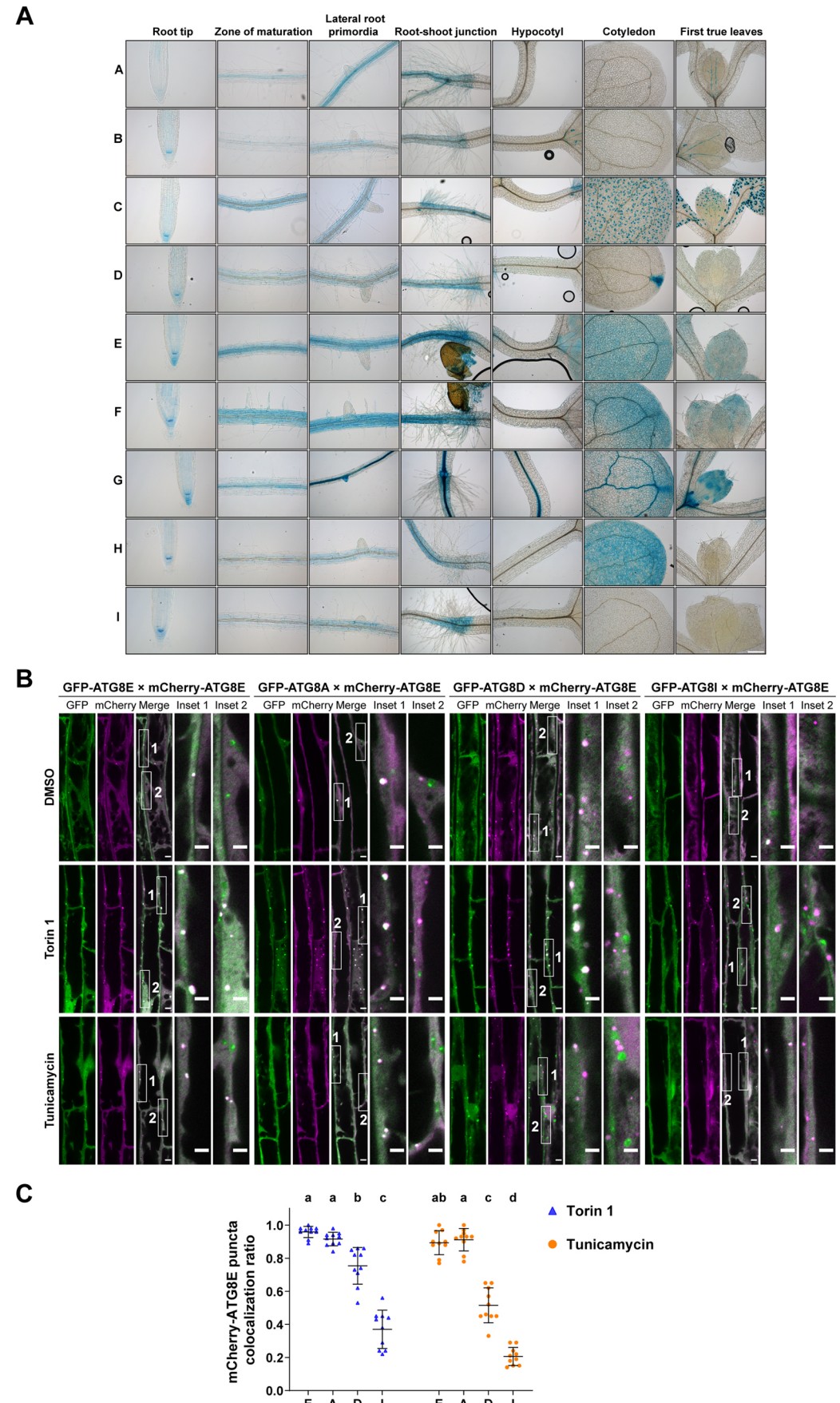

**Fig. 1.** See next page for legend.

**Fig. 1. *Arabidopsis thaliana* ATG8 isoforms exhibit tissue-specific expression patterns and form distinct autophagosomes within root cells.** (A) Representative GUS staining images showing the spatial-temporal expression patterns of nine *Arabidopsis* ATG8 isoforms. 10-day-old *Arabidopsis* seedlings expressing pATG8X::GFP-GUS (X represents the nine ATG8 isoforms, from A to I) were stained with GUS staining buffer. Scale bar: 100 μm. (B) Representative confocal microscopic images showing the colocalization of mCherry–ATG8E with different GFP–ATG8 isoforms in *Arabidopsis* root epidermal cells. 5-day-old *Arabidopsis* seedlings co-expressing mCherry–ATG8E with GFP–ATG8E, GFP–ATG8A, GFP–ATG8D or GFP–ATG8I were incubated in ½ MS liquid medium containing either DMSO (as mock condition) for 2 h, 5 μM Torin 1 for 2 h, or 10 μg/ml tunicamycin for 4 h. Representative images of 10 replicates are shown here for A and B. Scale bars: 5 μm (main images); 3 μm (magnifications). (C) Quantification of mCherry–ATG8E colocalization ratio for the *Arabidopsis* root epidermal cells imaged in B. The mCherry–ATG8E colocalization ratio was calculated as the ratio of the number of mCherry–ATG8E puncta that colocalized with GFP–ATG8 isoforms to the total number of mCherry–ATG8E puncta. Bars indicate the mean±s.d. of 10 replicates. Brown–Forsy and Welch ANOVA tests with Dunnett's T3 multiple comparisons tests were used for statistically comparing the colocalization difference between each treatment group, with groups different from other groups at $P<0.05$ indicated by letters.

representing both clades. First, we analyzed the complemented lines with confocal microscopy. Both GFP–ATG8A and GFP–ATG8H formed bright cytoplasmic puncta, which further accumulated in the vacuole upon concanamycin A treatment and underwent autophagic flux (Fig. S2A,B). The expression levels of GFP–ATG8A and GFP–ATG8H were measured via RT-qPCR, to ensure comparability of the two complementation lines (Fig. S2C).

To assess to what extent they were able to recover the autophagic function and evaluate potential isoform-specific responses to different stressors, we subjected the two complementation lines to starvation assays. Upon carbon starvation, both complementation mutants exhibited a similar phenotype to Col-0 (Fig. 4A). This suggests that both ATG8A and ATG8H can mediate the autophagic recycling of cellular material that ensures survival during carbon deprivation. In contrast, during nitrogen starvation only GFP–ATG8A expression recovered the sensitivity to nitrogen starvation. GFP–ATG8H-expressing lines were similar to the *Δatg8* mutant (Fig. 4B). These results provide functional genetic evidence for ATG8 specialization in *Arabidopsis thaliana*.

To support these findings, we performed autophagic flux assays under carbon and nitrogen starvation. Under carbon starvation conditions, both *Δatg8/+GFP-ATG8A* and *Δatg8/+GFP-ATG8H* had similar NBR1 flux, in contrast to the *Δatg8* mutant (Fig. 4C). This is consistent with the phenotyping results and indicates that both ATG8A and ATG8H can trigger bulk autophagy in response to carbon deprivation. However, during nitrogen starvation NBR1 flux in *Δatg8/+GFP-ATG8H* was similar to the *Δatg8* mutant (Fig. 4D), further corroborating the hypothesis that ATG8H, unlike ATG8A, is not able to fully operate autophagy in response to nitrogen deprivation. This is in line with a recent study that used AtATG8I and demonstrated that it is unable to complement nitrogen starvation sensitivity (Zou et al., 2025). Altogether, these results suggest Clade II isoforms are insufficient to carry out the recycling processes required for tolerating nitrogen starvation.

### ATG8A and ATG8H have distinct proximal interactomes during nitrogen starvation

Following the observation that ATG8A and ATG8H do not respond equally to nitrogen starvation stress, we reasoned that the two isoforms might be interacting with different proteins that are involved in autophagy signaling or cargo recognition. Indeed, our results could be explained by the inability of ATG8H to engage

with the nitrogen starvation response signaling or its failure to associate with the selective autophagy receptors recognizing the cargoes that need to be degraded to cope with nitrogen deprivation. To test these hypotheses, we complemented the *Δatg8* mutant with ATG8A and ATG8H fused to the biotinylating enzyme TurboID and determined the ATG8 proximal interactomes during nitrogen starvation (Fig. 4M). We used TurboID alone as negative control. 47 proteins exhibited specific association with both or only one of the two ATG8 isoforms (Table S3). Among these, numerous well-known interactors were found, including several ATG proteins. Interestingly, whereas some ATG proteins showed similar levels of association with ATG8A and ATG8H, such as ATG3, ATG7, ATG14B and ATG18F, other ATG proteins appear to interact prevalently or exclusively with just one isoform (Table S3). In the case of ATG1, ATG1A seems to interact with ATG8A uniquely, whereas ATG1B could associate with both isoforms but still exhibited a stronger association with ATG8A in the conditions tested. ATG1 kinase initiates autophagosome biogenesis and constitutes one of the major targets for autophagy regulation (Kamada et al., 2010; Mizushima, 2010). In light of this, our results might indicate that, during nitrogen starvation, ATG1A and ATG1B recruit ATG8A preferentially to promote nutrient replenishment. It is also interesting to observe that less than one-third of the total ATG8 interactors are shared between ATG8A and ATG8H (Fig. 4N), whereas 22 proteins specifically interact with ATG8A, and eight proteins interact with only ATG8H. As a proof of concept, our proximity labeling analysis suggested that the adaptor protein CFS1 specifically interacts with ATG8A but not with ATG8H, consistent with our previous results (Zhao et al., 2022). Although other interactors need to be further validated, these results suggest single-isoform TurboID lines provide an effective tool to study ATG8 specialization in a wide range of stress conditions.

### DISCUSSION

Given that the core autophagy machinery is shared across various selective autophagy pathways, the question of how cells achieve subcellular compartmentalization of these concurrent mechanisms remains unresolved. A plausible explanation is ATG8 isoform specialization, wherein distinct ATG8 variants interact with specific adaptors, receptors or ATG proteins to direct and compartmentalize autophagic processes. Previous biochemical and proteomic analyses in potato have supported this hypothesis, revealing isoform-specific interactomes (Zess et al., 2019). However, the co-occurrence of multiple ATG8 isoforms within individual autophagosomes (Fig. 1) suggests that some unique interactors might have been overlooked.

In this study, we provide genetic evidence for ATG8 specialization in plants using an *Arabidopsis Δatg8* nonuple mutant complemented with individual ATG8 isoforms. Where mitophagy is observed in HeLa cells lacking multiple ATG8 genes (Nguyen et al., 2016), the *Arabidopsis Δatg8* mutant failed to perform mitophagy and pexophagy, as evidenced by the absence of mitophagosomes and the accumulation of autophagy substrates (Fig. 3). Although double-membraned vesicles were observed in *Δatg8* cells, these are likely provacuolar compartments, which are independent of autophagy.

Interestingly, complementation experiments revealed functional divergence among ATG8 isoforms. Whereas both ATG8A and ATG8H restored carbon starvation sensitivity, only ATG8A was able to complement nitrogen starvation sensitivity (Fig. 4). Consistent with their specialization, proximity labeling proteomics demonstrated distinct interactomes for ATG8A and ATG8H. Further studies are

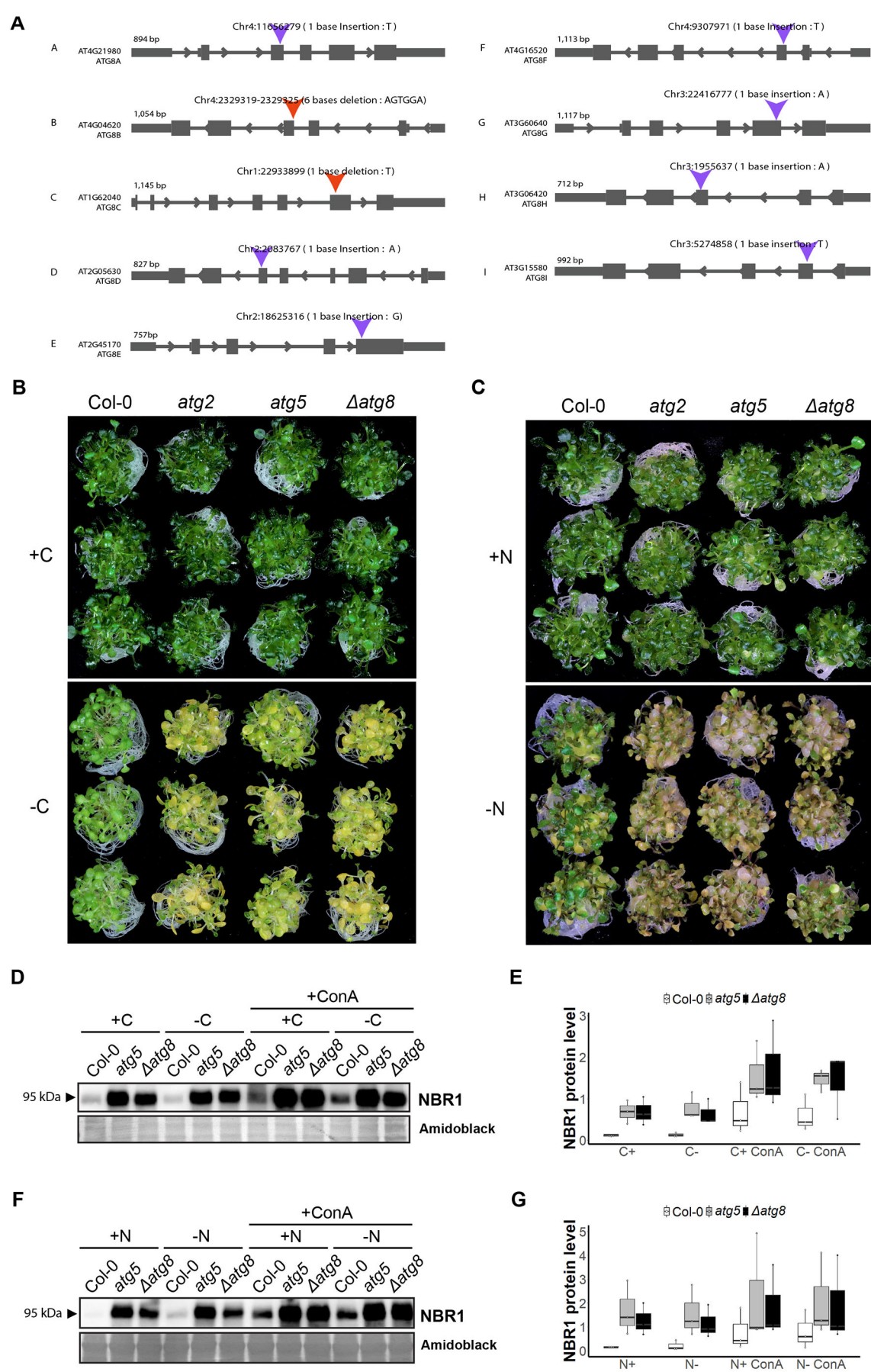

**Fig. 2.** See next page for legend.

**Fig. 2. The *atg8* nonuple mutant Δ*atg8* is deficient in autophagy.**
(A) Schematic representation of gene models illustrating CRISPR-induced mutations in the ATG8 gene. Blue arrows denote insertion sites; red arrows indicate deletion sites. The forward and reverse arrows within the gene model indicate the directionality of gene transcription, reflecting the strandedness of the gene. (B,C) Carbon (B) and nitrogen (C) starvation phenotypic assays comparing Col-0, *atg5* and Δ*atg8* (*n*=3) in carbon-rich (+C) or carbon-deficient (−C) ½ MS liquid medium and nitrogen-rich (+N) or nitrogen-deficient (−N) ½ MS liquid medium. (D–F) Western blots comparing endogenous NBR1 levels in Col-0, *atg5* and Δ*atg8* upon carbon (D) and nitrogen (F) starvation, in combination with concanamycin A (1 μM). Relative quantification of protein bands is reported in boxplots (E,G), representing the calculated values for three biological replicates. Boxes range from the 25% to the 75% percentiles, the horizontal lines correspond to the median and whiskers indicate the 5% and 95% percentiles.

necessary to link the nitrogen sensitivity phenotype to the differentially interacting proteins. Nevertheless, our findings establish the functional specialization of ATG8 isoforms in plants, providing a framework for understanding how cells fine-tune autophagic processes in response to diverse and overlapping signals.

## MATERIALS AND METHODS
### Plant material and cloning
All *Arabidopsis thaliana* lines used originate from the Columbia (Col-0) ecotype. The Δ*atg8* mutant was generated with two rounds of CRISPR editing. First, the CRISPR/Cas9 construct including gRNAs for all nine AtATG8 genes and an intronized Cas9 was assembled according to a protocol previously described (Grützner et al., 2021). The construct was assembled in binary vector pAGM62636 (Grützner et al., 2021), which contains a p15 origin of replication for low copy number replication in *E. coli* and an *Agrobacterium rhizogenes* A4 origin for single copy replication in Agrobacterium, resulting in plasmid pAGM70811. This vector backbone was chosen to minimize the risk or recombination between the 9 guides RNA cassettes present on the same plasmid. The gRNAs sequences (5′-3′) are as follows: ATG8A (AT4G21980), AGCTTAC GGGAATTCTGTCA; ATG8B (AT4G04620): GAACTCAATACAGGT-GATTG; ATG8C (AT1G62040), CAGTTAGATCAGCTGGAACA; AT G8D (AT2G05630), AAGAGGATGTTCATGCTTG; ATG8E (AT2G 45170), CTGTTAGGTCTGATGGCACA; ATG8F (AT4G16520), TTCA-GAGAAGAGAAGGGCAG; ATG8G (AT3G60640), GAGGAGACAGT ACCGGTGGG; ATG8H (AT3G06420), AAACGCAGATCTGCCAGACA; and ATG8I (AT3G15580), GATGAAAGGCTCGCGGAGTCG.

Primary transformants were genotyped by amplicon sequencing. The data was analyzed using a custom-built pipeline to retrieve indel frequencies in the ATG8 genes, using bwa and samtools *mpileup* (Xing et al., 2014; Lampropoulos et al., 2013). Code and documentation are available at https://github.com/pierre-bourguet/CRISPR_genotyping.git. Following sequencing, we could only confirm the successful mutation of eight of the nine isoforms. Therefore, a second CRISPR/Cas9 was employed to target the last gene (ATG8D; AT2G05630). The construct was assembled according to the protocol previously described (Langmead and Salzberg, 2012), on pHEE401E plasmid, with the following gRNAs sequences (5′-3′): GAACAACAGAGACTCGACCA and GTGATGTCCCGGATA-TTGAT. The nonuple mutant Δ*atg8* contains both T-DNA CRISPR/Cas9 cassettes and is resistant to BASTA and hygromycin. The mutations of nine ATG8 isoforms were confirmed via single-cell sequencing.

All the plasmids, except pAGM7081, were assembled through the GreenGate cloning procedure (Danecek et al., 2021) and were constructed as follows: pGGZ003_ATG8X::GFP-GUS (where X represents the nine ATG8 isoforms, from A to I); pUBQ10::mCherry-ATG8E; pUBQ10:: GFP-ATG8A; pUBQ10::GFP-ATG8D; pUBQ10::GFP-ATG8E; pUBQ10:: GFP-ATG8I; pGGSUN_RPS5::mCherry-TurboID; pGGSUN_RPS5:: mCherry-TurboIDATG8A; pGGSUN_RPS5::mCherry-TurboID-ATG8H; pGGSUN_HTR5::GFP-ATG8A; pGGSUN_HTR5::GFP-ATG8H. Apart from pATG8X::GFP-GUS-expressing plants, which are hygromycin resistant, transformants were selected via seed coat fluorescence. The coding sequences of ATG8A and ATG8H carry silent mutations to avoid

CRISPR/Cas9 targeting. The point mutations from the start codon are the following: ATG8A, 81 bp T>A, 84 bp C>T, 87 bp A>G, 93 bp C>G; for ATG8H, 123 bp C>T, 126 bp A>C, 129 bp T>C, 135 bp A>G, 138 bp C>T.

### DNA sequencing and analysis
High-quality DNA was extracted using the cetyltrimethylammonium bromide (CTAB) method and used to construct Illumina-compatible libraries with the Nextera XT DNA Library Preparation Kit, following the manufacturer's instructions. Sequencing was performed on an Illumina NextSeq instrument in paired-end 150 bp mode. Raw FASTQ files obtained from sequencing were quality-checked and adapter-trimmed using TrimGalore (https://github.com/FelixKrueger/TrimGalore) with default settings. The trimmed FASTQ files were aligned to the TAIR10 genome using Bowtie2 (Langmead and Salzberg, 2012) with the parameters -D 15 -R 2 -N 0 -L 22 -i S,1,1.15. The resulting aligned BAM files were sorted and indexed using SAMtools (Danecek et al., 2021) and manually inspected for insertions or deletions using the Integrative Genomics Viewer (IGV). Deletions and insertions were manually inspected, and the corresponding changes in cDNA and protein sequences were catalogued (Table S1).

### Plant growth and treatments
For standard plant growth, *Arabidopsis* seeds were gas sterilized with sodium hypochlorite+HCl (10:1 v/v), sown on water-saturated soil and grown in 16-h-light–8-h-dark photoperiod with 165 μmol m$^2$ s$^{−1}$ light intensity. For *in vitro* growth, *Arabidopsis* seeds were surface sterilized in 70% ethanol for 10 min twice, then rinsed in absolute ethanol and dried on sterile paper. Seeds were sown on ½ MS liquid medium [Murashige and Skoog salt+Gamborg B5 vitamin mixture (Duchefa) supplemented with 0.5 g/liter MES and 1% sucrose, pH 5.7], vernalized at 4°C in the dark for 2 days, and then grown under LEDs with 85 μM/m²/s with a 14-h-light–10-h-dark photoperiod.

For drug treatments, all drugs were dissolved in DMSO and added to the desired concentration: 3 μM Torin 1 (Santa Cruz Biotechnology; CAS 1222998-36-8), 10 μg/ml tunicamycin (Santa Cruz Biotechnology; CAS 11089-65-9), 1–2 μM concanamycin A (Santa Cruz Biotechnology; CAS 80890-47-7), 50 μM DNP (Sigma-Aldrich; D198501-1KG). An equal amount of pure DMSO was added to control samples.

For confocal microscopy, *Arabidopsis* seeds were sterilized by 70% ethanol+0.05% Tween 20 (Sigma-Aldrich) for 5 min and were subsequently by 100% ethanol for 10 min. Sterilized seeds were stored in sterile water at 4°C for 1 day for vernalization. Vernalized seeds were spread on ½ MS medium plates (+1% plant agar; Duchefa) and vertically grown at 21°C at 60% humidity under LEDs with 50 mM/m²s a and a 16-h-light–8-h-dark photoperiod. 5-day-old seedlings were incubated in ½ MS medium containing DMSO for 2 h, 3 μM Torin 1 for 2 h, 10 μg/ml tunicamycin for 4 h or 2 μM concanamycin for 2.5 h before imaging.

### Carbon and nitrogen starvation assays
*A. thaliana* seeds (~30 per sample, three replicates per condition) were sterilized with ethanol, sown in ½ MS liquid medium, vernalized at 4°C in the dark for 2 days and grown at 21°C under LEDs with 85 μM/m²/s and a 14-h-light–10-h-dark photoperiod. The starvation treatments were performed on 9-day-old seedlings, by replacing ½ MS liquid medium with the same medium (as control), ½ MS liquid medium without sucrose for carbon starvation or ½ MS liquid medium without nitrogen (Murashige & Skoog without Nitrogen, Caisson Laboratories; MSP21) for nitrogen starvation. Prior to medium replacement, the seedlings were washed twice with 1 ml of the new medium to ensure proper removal of the previous medium. For carbon starvation, the seedlings were kept in the dark. Pictures of the samples were taken after 4 days of carbon starvation and 6 days of nitrogen starvation.

### Confocal microscopy
For confocal microscopy, *Arabidopsis* seedlings were placed on a microscope slide with water and covered with a coverslip. The epidermal cells of root transition and elongation zone were used for image acquisition. Confocal images were acquired via an upright point laser scanning confocal microscope ZEISS LSM800 Axio Imager.Z2 (Carl Zeiss) equipped with

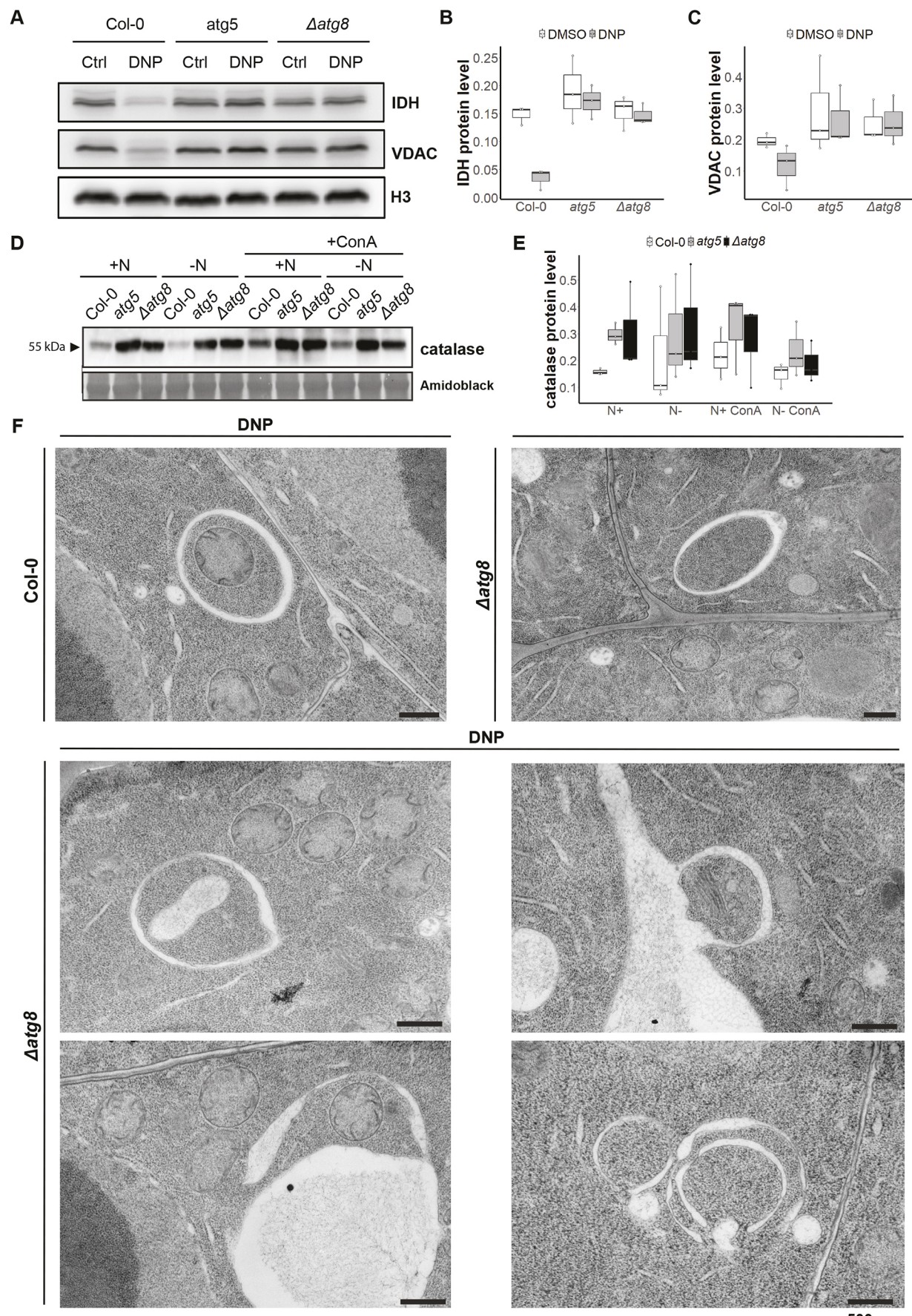

**Fig. 3.** See next page for legend.

**Fig. 3. The *Δatg8* mutant is not able to perform mitophagy and pexophagy.** (A) Western blots comparing endogenous IDH and VDAC levels in Col-0, *atg5* and *Δatg8* upon DNP treatment. (B,C) Relative protein levels of IDH (B) and VDAC (C) are represented in boxplots reporting the calculated values of three biological replicates. (D) Western blots comparing endogenous catalase levels in Col-0, *atg5* and *Δatg8* upon nitrogen starvation treatment (−N). Nitrogen-rich medium, +N. (E) Relative catalase protein levels, calculated from three biological replicates. (F) Electron micrographs of Col-0 and *Δatg8* root cells treated with DNP or DMSO. Images representative of more than 30 sections from three individual roots. Scale bars: 500 nm. For all boxplots, the box represents the 25–75th percentiles, and the median is indicated. The whiskers show the 5th–95th percentiles.

high-sensitive GaAsP detectors (gallium arsenide), a LD C-Apochromat 40× objective lens (numerical aperture 1.1, water immersion) and ZEN software (blue edition 3.8, Carl Zeiss). GFP signals were excited at 488 nm and detected between 488 and 545 nm. mCherry signals were excited at 561 nm and detected between 570 and 617 nm.

### Image processing and statistics
Confocal images were processed and quantified by Fiji (version 1.52, Fiji). The mCherry–ATG8E colocalization ratio was calculated as the ratio of the number of mCherry–ATG8E puncta that colocalized with GFP-ATG8 isoforms to the total number of mCherry–ATG8E puncta. Statistics tests were performed via GraphPad Prism 8.1.1.

### Protein extraction and western blotting
20-40 *A. thaliana* seeds per sample were surface sterilized with ethanol, sown in ½ MS liquid medium, vernalized at 4°C in the dark for 2 days and grown for 7 days at 21°C under LEDs with 85 μM/m²/s with a 14-h-light–10-h-dark photoperiod. For starvation treatments, performed overnight, the ½ MS liquid medium was replaced with the same medium (as control), ½ MS liquid medium without sucrose for carbon starvation or ½ MS liquid medium without nitrogen (Murashige & Skoog without Nitrogen, Caisson Laboratories; MSP21) for nitrogen starvation. For carbon starvation, the samples were kept in the dark. When required, 1 μM concanamycin A (Santa Cruz Biotechnology; CAS 80890-47-7) was added to the new medium. The seedlings were harvested in safe lock Eppendorf tubes containing 2 mm diameter glass beads, flash frozen in liquid nitrogen and pulverized using a Silamat S7 (Ivoclar vivident). Total proteins were extracted in 2× Laemmli buffer by agitating the samples in the Silamat S7 for 20 s. The samples were boiled at 70°C and 1000 rpm shaking for 10 min, then centrifuged at maximum speed (21,300 *g* for 10 min) with a benchtop centrifuge. Total proteins were quantified with the Amido Black method. 10 μl of supernatant was added to 190 μl of deionized water, vortexed and then mixed with 1 ml of Amido Black buffer [10% acetic acid, 90% methanol, 0.05% (w/v) Amido Black (Napthol Blue Black, Sigma N3393)] by inverting the tubes. After 10 min centrifugation at maximum speed, pellets were washed with 1 ml of wash buffer (10% acetic acid, 90% ethanol), mixed by inversion, and centrifuged for another 10 min at maximum speed. Pellets were resuspended in 0.2 N NaOH and $OD_{630\,nm}$ was measured, with NaOH solution as blank, to quantify protein concentration with the OD=a[C]+b determined curve. 15 μg of total protein extracts were separated on SDS-PAGE gels and blotted onto PVDF Immobilon-P membrane (Millipore). NBR1 was detected using the anti-NBR1 antibody (rabbit polyclonal; Agrisera; AS14 2805) diluted 1:10,000. Catalase proteins were detected with Anti-cat antibody (rabbit polyclonal; Agrisera; AS09 501), diluted 1:1000. GFP was detected with the anti-GFP antibody (mouse monoclonal; Roche; 11814460001), diluted 1:5000. mCherry was detected using the anti-RFP antibody [6G6] (mouse monoclonal; ChromoTek; AB_2631395), diluted 1:5000. ATG8 was detected with anti-ATG8(a-i) antibody (rabbit polyclonal; Agrisera; AS14 2811), diluted 1:4000. Rabbit polyclonal antibody was detected with a goat anti-rabbit IgG HRP-linked antibody (Invitrogen, 65-6120) diluted 1:5000. Mouse monoclonal antibody was detected with a goat anti-mouse IgG HRP-linked antibody (Bio-Rad, #1706516) diluted 1:5000. Hybridized membranes were reacted with SuperSignal™ West Pico PLUS Chemiluminescent Substrate (Thermo Fisher Scientific) and imaged using an iBright CL1500 Imaging System (Invitrogen).

Protein bands were quantified with ImageJ according to the protocol previously described (Stael et al., 2022) and normalized on the loading control. The calculated values for three biological replicates are presented in boxplots, where the boxes range from the 25% to the 75% percentiles, the horizontal lines correspond to medians and whiskers indicate the 5% and 95% percentiles. Boxplots and other visualizations were generated using the ggplot2 package in R (https://ggplot2.tidyverse.org). For mitophagy assays, 5-day-old *Arabidopsis* seedlings were treated with 50 μM DNP (Sigma-Aldrich, D198501-1KG), or an equal amount of DMSO, for 2–3 h in the dark, then moved to liquid ½ MS medium for 1 h recovery under light. Protein extraction and immunoblot analysis was performed as previously reported (Ma et al., 2021).

### GUS staining
10-day-old *Arabidopsis* seedlings expressing pATG8X::GFP-GUS (X represents the nine ATG8 isoforms, from A to I) were first immersed in acetone for 20 min and then washed with the GUS buffer (50 mM NaPO₄, 2 mM K-ferrocyanide, 2 mM K-ferricyanide, 0.2% Triton X-100). The washed samples were subsequently incubated in GUS staining buffer [GUS buffer+2 mM X-Gluc (Thermo Scientific)] under 37°C until a blue coloration was visible. The stained samples were then washed and discolored with 100% ethanol and were ready for photographing.

### RNA extraction and RT-qPCR
20–40 *A. thaliana* seeds for each sample were surface sterilized with ethanol, sown in ½ MS liquid medium, vernalized at 4°C in the dark for 2 days and grown at 21°C under LEDs with 85 μM/m²/s with a 14-h-light–10-h-dark photoperiod. 7-days old seedlings were harvested in safe lock Eppendorf tubes containing 2 mm diameter glass bead, flash frozen in liquid nitrogen and ground using a Silamat S7 (Ivoclar vivident). RNA was extracted using the Direct-zol™ RNA MiniPrep Kit (Zymo Research) according to the protocol. 1 μg of total RNA was treated with DNase I (Thermo Fisher Scientific), and one-third of the reaction was reverse transcribed using the SuperScript™ IV First-Strand Synthesis System (Invitrogen) with 2.5 μM oligo d(T)20 primer. RT-qPCR was performed in 10 μl with Kapa SYBR Fast qPCR universal mix (Kapa Biosystems) using 200 nM final primer concentration in a Roche LightCycler 96 with the program: (95°C, 5 min) initial denaturation; (95°C 10 s, 60°C 15 s, 72°C 15 s) cycling for 45 times, with single acquisition mode. Melting curve: 95°C 5 s, 55° 1 min, 95°C in continuous acquisition mode with a ramp rate of 0.11°C/s, 5 acquisition per °C. For each sample, three technical replicates were prepared, and expression levels were normalized on ACT2 (AT3G18780) and UBQ9 (AT5G37640) as internal controls. Primers were designed according to the instructions in the protocol and are as follows (5′-3′): GGTCCTGCTGGAGTTCGTG+GAAGATTCACTCATCCTTGCCT-CGAG (for GFP–ATG8A); GGTCCTGCTGGAGTTCGTG+CTCTCAT-CAGAGGAGAATTGATCCTTGAA (for GFP–ATG8H); ATCGTGTGTG-ACAATGGTACCG+GGTTCATCCCAACCATGACAC (for ACT2); and TCAGCACATTCCAGCAGATGT+ATTCATAAAACCCCAGCTTTTTA-AGCCT (for UBQ9).

### Affinity purification of biotinylated proteins and nanoLC-MS/MS analysis
*A. thaliana* seeds were surface sterilized with ethanol, stratified for 2 days at 4°C in the dark and then grown in ½ MS (Duchefa), 0.5% MES and 1% sucrose under LEDs with 85 μM/m²/s and a 14-h-light–10-h-dark photoperiod. 7-day-old seedlings were washed and treated with nitrogen-deficient ½ MS medium (or control ½ MS medium) overnight and the following morning 50 μM biotin was added to the medium. After 1 h of biotin incubation, the seedlings were quickly rinsed in ice-cold water, dried and frozen in liquid nitrogen. Around 1 g of plant tissue was used for each sample and the affinity purification of biotinylated proteins was performed as previously described (Mair et al., 2019).

For MS analysis, the nano HPLC system (UltiMate 3000 RSLC nano system) was coupled to an Orbitrap Exploris 480 mass spectrometer, equipped with a Nanospray Flex ion source (all parts Thermo Fisher Scientific). Peptides were loaded onto a trap column (PepMap Acclaim C18, 5 mm×300 μm ID, 5 μm particles, 100 Å pore size, Thermo Fisher

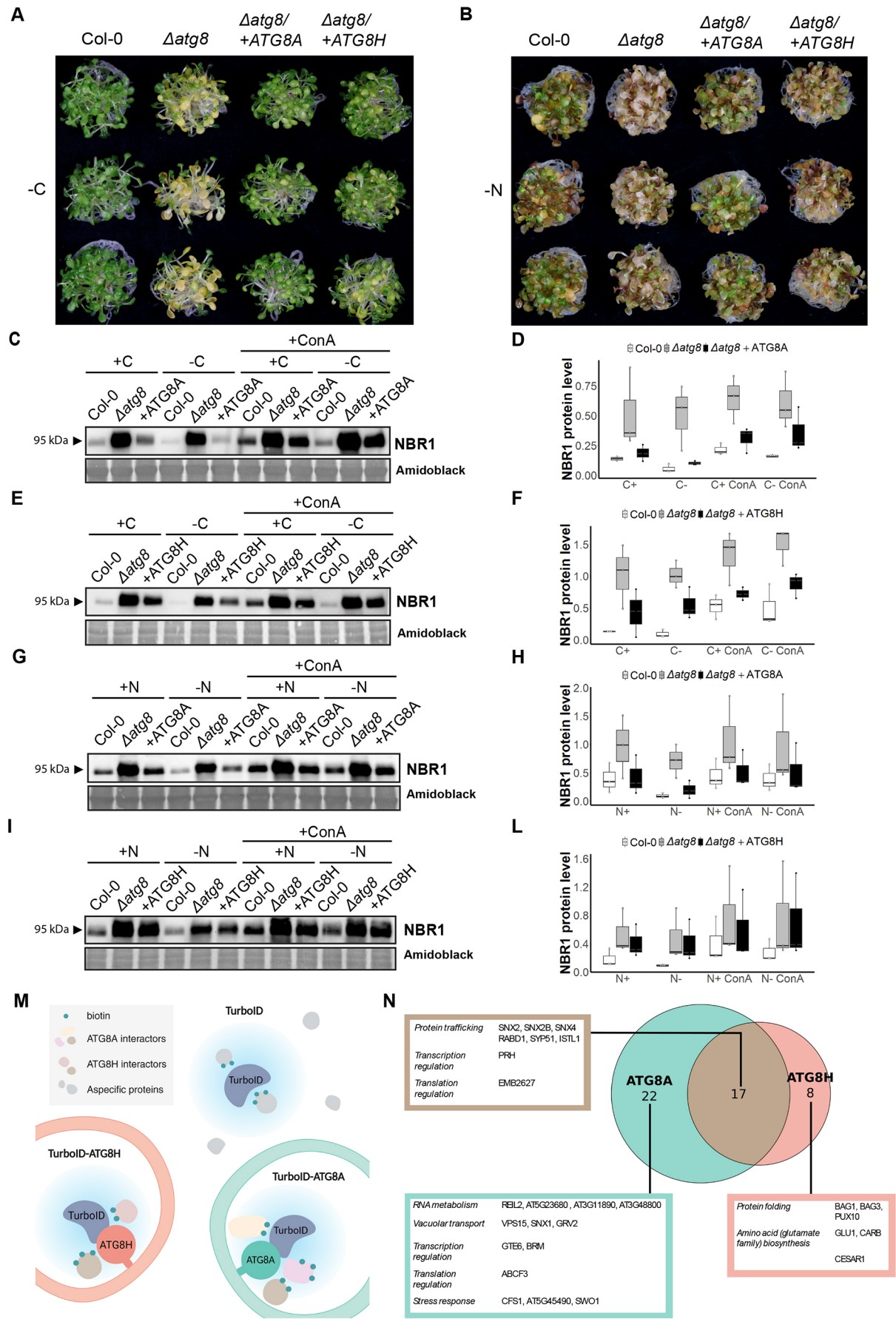

**Fig. 4.** See next page for legend.

**Fig. 4. Complementation of *Δatg8* with ATG8A or ATG8H reveals functional specialization of ATG8 isoforms.** (A,B) Carbon (A) and nitrogen (B) starvation phenotypic assays comparing Col-0, *Δatg8* and complementation lines *Δatg8* /+ATG8A, *Δatg8*/+ATG8H (*n*=3) in carbon-deficient (−C) ½ MS liquid medium and nitrogen-deficient (−N) ½ MS liquid medium. Images representative of two repeats. (C,E,G,I) Western blots comparing endogenous NBR1 levels in Col-0, *Δatg8*, *Δatg8*/+ATG8A (C,G) and *Δatg8* /+ATG8H (E,I) upon carbon (C,E) and nitrogen (G,I) starvation, in combination with concanamycin A (1 µM). (D,F,H,L) Relative quantification of protein bands is reported in boxplots next to the corresponding western blot, representing the calculated values for three biological replicates. (M) Schematic representation of TurboID proximity labeling analysis. (M) Venn diagram reporting common and unique interactors of ATG8A and ATG8H under nitrogen starvation. For all boxplots, the box represents the 25–75th percentiles, and the median is indicated. The whiskers show the 5th–95th percentiles.

Scientific) at a flow rate of 25 µl/min using 0.1% TFA as mobile phase. After loading, the trap column was switched in line with the analytical column (PepMap Acclaim C18, 500 mm×75 µm ID, 2 µm, 100 Å, Thermo Fisher Scientific). Peptides were eluted using a flow rate of 230 nl/min, starting with the mobile phases 98% A (0.1% formic acid in water) and 2% B (80% acetonitrile, 0.1% formic acid) and linearly increasing to 35% B over the next 120 min. This was followed by a steep gradient to 95% B in 1 min, stayed there for 6 min and ramped down in 2 min to the starting conditions of 98% A and 2% B for equilibration at 30°C. The Orbitrap Exploris 480 mass spectrometer was operated in data-dependent mode, performing a full scan (*m/z* range 350–1200, resolution 60,000, normalized AGC target 300%) at three different compensation voltages (CV −45 V, −60 V and −75 V), followed by MS/MS scans of the most abundant ions for a cycle time of 0.9 s for each. MS/MS spectra were acquired using an isolation width of 1.2 *m/z*, normalized AGC target 200%, HCD collision energy of 30%, maximum injection time mode set to custom and resolution of 30,000. Precursor ions selected for fragmentation (include charge state 2–6) were excluded for 45 s. The monoisotopic precursor selection (MIPS) mode was set to peptide and the exclude isotopes feature was enabled.

## MS data processing

For peptide identification, the RAW files were loaded into Proteome Discoverer (version 2.5.0.400, Thermo Fisher Scientific). All MS/MS spectra were searched using MSAmanda v2.0.0.19924 (Dorfer et al., 2014). The peptide mass tolerance was set to ±10 ppm and fragment mass tolerance to ±10 ppm, the maximum number of missed cleavages was set to two, using tryptic enzymatic specificity without proline restriction. The RAW files were searched against the *Arabidopsis* database (32,785 sequences; 14,482,855 residues), supplemented with common contaminants and sequences of tagged proteins of interest. The following search parameters were used: oxidation on methionine, phosphorylation on serine, threonine and tyrosine, deamidation on asparagine and glutamine, iodoacetamide derivative on cysteine, β-methylthiolation on cysteine, biotinylation on lysine, ubiquitylation residue on lysine, ubiquitination on lysine, pyro-glu from q on peptide N-terminal glutamine, acetylation on protein N-terminus were set as variable modifications. The result was filtered to 1% false discovery rate (FDR) on protein level using the Percolator algorithm (Käll et al.. 2007) integrated in Proteome Discoverer. The localization of the post-translational modification sites within the peptides was performed with the tool ptmRS, based on the tool phosphoRS (Taus et al., 2011). Additionally, an Amanda score cut-off of at least 150 was applied. Protein areas have been computed in IMP-apQuant (Doblmann et al, 2019) by summing up unique and razor peptides. Resulting protein areas were normalized using iBAQ (Schwanhäusser et al., 2011) and sum normalization was applied for normalization between samples. Match-between-runs (MBR) was applied for peptides with high confident peak area that were identified by MS/MS spectra in at least one run. Proteins were filtered to be identified by a minimum of 2 PSMs in at least one sample and quantified proteins were filtered to contain at least three quantified peptide groups. Statistical significance of differentially expressed proteins was determined using limma (Smyth, 2004) (Table S2).

To identify potential interactors, the log2 fold change (log2FC) was calculated comparing average PSMs in TID-ATG8A samples (*n*=3) or TID-ATG8H samples (*n*=3) against the TID control (*n*=3). Proteins with average PSMs>3, log2FC>1 and *P*-value>0.05 were selected as potential interactors. The list of potential interactors is provided in Table S3. The mass spectrometry proteomics data have been deposited to the ProteomeXchange Consortium via the PRIDE partner repository with the dataset identifier PXD058764

## TEM

The TEM assay was performed following the previously established method (Ma et al., 2021; Kang, 2010). Briefly, 5-day-old *Arabidopsis* seedlings were germinated on ½ MS plates and dissected under microscopy before freezing. For high-pressure freezing, the root tips were collected and immediately frozen with a high-pressure freezer (EM ICE, Leica). For freeze substitution, the root tips were replaced with 2% osmium tetroxide in anhydrous acetone and maintained at −80°C for 24 h using an AFS2 temperature-controlling system (Leica). Subsequently, the samples were subjected to three washes with precooled acetone and slowly warmed up to room temperature over a 60-h period before being embedded in EPON resin. After resin polymerization, samples were mounted and trimmed. For the ultrastructure studies, 100 nm thin sections were prepared using an ultramicrotome (EM UC7, Leica) and examined with a Hitachi H-7650 TEM (Hitachi-High Technologies) operated at 80 kV. For each sample (wild-type and *Δatg8*), more than 30 sections from three individual roots were examined.

### Acknowledgements
We thank Vienna Biocenter Core Facilities (VBCF), particularly Proteomics, BioOptics and Plant Sciences. We thank the CLIP cluster (https://clip.science) for access for image analysis.

### Competing interests
The authors declare no competing or financial interests.

### Author contributions
Conceptualization: Y.D.; Data curation: A.D.C., N.G., R.K.P., P.B.; Formal analysis: A.D.C.; Funding acquisition: Y.D.; Investigation: J.Z., P.G., J.M.; Methodology: A.D.C., N.G., R.G.; Resources: N.G.; Supervision: B.-H.K., S.M., Y.D.; Visualization: A.D.C., J.Z., P.G., J.M., C.L., A.B.; Writing – original draft: A.D.C., N.G., R.K.P., Y.D.; Writing – review & editing: A.D.C., F.B., B.-H.K., S.M., Y.D.

### Funding
We acknowledge funding from Austrian Academy of Sciences, Austrian Science Fund (FWF, P32355, P34944, SFB F79, DOC 111), Vienna Science and Technology Fund (WWTF, LS17–047, LS21–009), and European Research Council (Project number: 101043370) and a Deutsche Forschungsgemeinschaft (DFG) Heisenberg Award. P.G. is supported by the Vienna International Postdoctoral Program (VIP2) and Marie Curie Fellowship (Project number: 847548). Open Access funding provided by Austrian Academy of Sciences. Deposited in PMC for immediate release.

### Data and resource availability
All the source data used to generate the main and supplementary figures have been deposited to Zenodo (doi:10.5281/zenodo.14277422). Genomic sequencing data generated in this study are deposited at the National Center for Biotechnology Information Gene Expression Omnibus (NCBI) under accession number GSE283481. The mass spectrometry proteomics data have been deposited to the ProteomeXchange Consortium via the PRIDE partner repository (ID: PXD058764).

### Peer review history
The peer review history is available online at https://journals.biologists.com/jcs/lookup/doi/10.1242/jcs.263803.reviewer-comments.pdf

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
