## [Peer Review File · Journal of Cell Science]

An *A.thaliana* mutant lacking all nine ATG8 isoforms provides genetic evidence for functional specialization of ATG8 in plants

Alessia Del Chiaro, Nenad Grujic, Jierui Zhao, Ranjith Kumar Papareddy, Peng Gao, Juncai Ma, Christian Lofke, Anuradha Bhattacharya, Ramona Gruetzner, Pierre Bourguet, Frédéric Berger, Byung-Ho Kang, Sylvestre Marillonnet and Yasin Dagdas
DOI: 10.1242/jcs.263803

Editor: Charlotte Kirchhelle

Review timeline

Original submission:	17 December 2024
Editorial decision:	17 February 2025
First revision received:	13 June 2025
Editorial decision:	8 July 2025
Second revision received:	29 July 2025
Accepted:	1 August 2025

Original submission

First decision letter

MS ID#: jcs.263803

MS TITLE: Nonuple atg8 mutant provides genetic evidence for functional specialization of ATG8 isoforms in *Arabidopsis thaliana*

AUTHORS: Alessia Del Chiaro; Nenad Grujic; Jierui Zhao; Ranjith K. Papareddy; Peng Gao; Juncai Ma; Christian Lofke; Anuradha Bhattacharya; Ramona Gruetzner; Pierre Bourguet; Frederic Berger; Byung-Ho Kang; Sylvestre Marillonnet; Yasin Dagdas

ARTICLE TYPE: Short Report

Dear Dr Dagdas,

We have now reached a decision on the above manuscript.

To see the reviewers' reports and a copy of this decision letter, please go to:

As you will see, the reviewers raise a number of criticisms. Some of these I find well-founded (in particular better characterisation of the mutant, which will be the mutant more useful as a tool for the community), others less so, particularly considering you have submitted this manuscript for a short report. If you think that you can deal satisfactorily with the criticisms on revision, I would be pleased to see a revised manuscript. We would then return it to the reviewers.

Reviewer 1

Although the autophagy machinery is highly conserved between major taxonomic groups, ATG8, a central autophagy gene, has undergone expansion uniquely in plants. This interesting short report provides convincing evidence to support the hypothesis that expansion of ATG8 underpins functional specialisation.

The authors successfully created a nonuple *atg8* mutant through two rounds of CRISPR-Cas9 genome editing and then tested complementation of different mutant phenotypes by representatives of two distinct ATG8 clades. The study employs well established assays to demonstrate that both ATG8A and ATG8H mediate autophagic recycling during carbon starvation but only 8A recovers sensitivity of the *atg8* nonuple mutant to nitrogen starvation. Extending this observation, proximity ligation suggests that the two ATG8 isoforms may interact with distinct groups of proteins. These interactions were not validated by an orthogonal technique, which would be required for a larger, more in-depth study but not in my opinion for a short report. The proximity data are in agreement with a study of potato ATG8 from some of the same authors (Zess et al., 2019). Whilst supportive, this detracts to some extent from the novelty of the current report. However, the manuscript answers a key question in the field and provides a valuable tool for plant autophagy research.

Comments for the author

TEM was used to demonstrate lack of mitophagy in the *atg8* nonuple mutant, with representative images presented. However, I could not find details of how many sections were examined and how prevalent the mitophagosomes are in wild type. It would be helpful to provide details of the sampling and any quantification strategy to give confidence in the result.

The effect of ATG8H is not as marked as that for ATG8A in Fig. 4C; perhaps this could be mentioned in the text?

Please provide a reference for mitophagy in HeLa cells lacking ATG8 (bottom of page 6).

Reviewer 2

Advance summary and potential significance to field

The manuscript entitled "Nonuple *atg8* mutant provides genetic evidence for functional specialization of ATG8 isoforms in *Arabidopsis thaliana*" describes the functional diversification of the nine ATG8 homologs in *Arabidopsis* by generating a complete knock-out mutant of ATG8. Their results show that *Arabidopsis* ATG8 homologs are differentially expressed and could have functional diversification. The generated nonuple *atg8* KO mutant will be an important tool for the whole community and the findings described in this manuscript are of interest to a wide range of plant scientists.

Comments for the author

The manuscript is very well written and I have only a few comments, mainly regarding the nonuple *atg8* mutant. For its use in future research, it would be important that some aspects of the mutant are described in detail.

1. Figure 1A: The paper by Thompson et al (DOI: 10.1104/pp.105.060673) also investigated the differential induction of ATG8 homologs. In addition, a previous review by Woltering et al (DOI: 10.3389/fpls.2020.579875) describes the differences in mRNA and protein abundance of ATG8 homologs based on expression and proteomics data and Chung et al (doi: 10.1104/pp.108.126714) investigates the ATG8 homologs in maize. These papers should be cited for discussing the results shown here.

2. Figure 1B: I may have missed the description, but it was not clear to me under which promoter the GFP/mCherry ATG8 homologs were expressed. The methods only describe pGGSUN_HTR5::GFP-ATG8A and pGGSUN_HTR5::GFP-ATG8H. If non-endogenous promoters are used, a colocalization assay may not reflect physiological situations due to different expression patterns. The authors should describe the experimental set-up and may wish to comment on this.

3. Fig. 2A and in general to the nonuple *atg8* mutant: Although the data indicate that the generated nonuple mutant is a complete knock-out mutant, some additional information should be provided. It would be helpful if the authors described the translational consequences of the indel events for each homolog. For ATG8B, the 6 bp deletion that does not cause a frameshift and generate a premature stop codon. Here, it is not clear whether a 2 amino acid deletion affects ATG8B function. If ATG8B is still intact, the *atg8* mutant is an octuple instead of a nonuple mutant.

This doesn't diminish the importance of this genetic material or the impact of this study, but it should be clarified for future use. Can the authors map the position of the deletion in the protein structure and speculate? qPCR should be performed to investigate whether ATG8B transcripts are still present in the mutant as non-functional mRNA would decay.

4. Figure 3C and page 4, line 56 to page 5, line 9/page7 lines 1-2 "... we did not observe any mitophagosomes in $\Delta atg8$ mutant (Fig. 3C). However, we could still detect double-membrane structures that resemble autophagosomes in the $\Delta atg8$ mutant." These sentences are a bit too speculative. It has not been investigated in plants whether and to what extent phagophores or even autophagosomes (without ATG8) can be formed. Although more specialised in function, ATG8-independent autophagy could still occur. The localisation of non-ATG8 autophagosomal markers in the $atg8$ mutant and provacuole markers should be tested to clarify this. If this is not possible, the sentence should at least be rephrased so that the formation of autophagosomes is not categorically excluded.

5. Figure 4A/B: The expression levels of ATG8A and ATG8H should be examined. Otherwise, it cannot be excluded that the difference in complementation of the $\Delta atg8$ mutant simply reflects the level of expression rather than the differences in the two homologs.

6. Figure 4D: The presentation makes it difficult to see the differences between carbon and nitrogen starvation. The authors may consider presenting the quantification separately, if possible, by including quantification of all replicates.

7. Figure 4E. The Turbo-ID data are highly interesting but would need further validation to be truly informative. The authors may even want to consider removing this data from the current manuscript as the main conclusions seems mostly well supported by the rest of the data. If they decide to include the data, it is understandable, but the experiment should be described a little more in detail. Are the expression levels of the Turbo-ID constructs for ATG8H and ATG8A similar? Can at least some of the "differential" interactor candidates be verified?

Reviewer 3

Advance summary and potential significance to field

In this study, the authors have generated transcriptional and translational ATG8 reporters in *Arabidopsis thaliana* to reveal their expression and function diversity, which has not been addressed for a long time.

Comments for the author

1. Using nonuple $\hat{a}t g 8$ mutant, the authors showed that it is hypersensitivity to carbon and nitrogen starvation. However, in a recent publication (<https://doi.org/10.1038/s41467-024-55754-1>), this group has used the same mutant to demonstrate its sensitivity to nitrogen starvation and the functional diversity between ATG8f and ATG8i, which corresponding to ATG8A and ATG8H used in this study. They have claimed that *Arabidopsis* ATG8 isoforms differ in their capacity to carry out autophagic flux. The authors need to cite this study and remove the overlapping result or to supplementary data.

2. Figure titles need further editing to describe concisely. For example, 'Supplementary Figure 2. *Arabidopsis* ATG8 complementation lines have normal autophagosome structure and autophagic flux under control and nutrient-deficient conditions.'
If they are normal, the overall conclusion for functional specification is incorrect.

"The $\hat{a}t g 8$ mutant is not able to perform mitophagy and pexophagy."
In comparison with $atg5$ mutant, it seemed that $atg8$ mutant contains less IDH, VDAC and catalase after treatment. Better to quantify the immunoblot result.

3. To my understanding, functional specialization might be related to autophagosome formation, which has been reported for human ATG8 orthologs, from expansion, maturation to

completion. Certainly, it seemed that the proximitomes for ATG8A and ATG8I are different, but there is no verification for any candidates, at least one example can be tested. Though it seemed that not all ATG8 isoforms overlaps with ATG8e, but whether they all have similar expression level as ATG8e?

4. As there are still some puncta colocalized with ATG8e, real time imaging to show whether they are sequentially recruited to the autophagosome will be useful to provide more information for their functional specialization during autophagosome formation. Otherwise, it is possible that the autophagosome has already been decorated enough ATG8 proteins.

5. For the complementation lines, both ATG8A and ATG8i were driven by the strong HTR5 promoter. As shown in Figure 1A, different isoforms showed various expression patterns in plant tissues. How to explain on the medium with carbon, GFP-ATG8A and GFP-ATG8i both were already degraded?

6. For the *atg8a-i* mutant, as ATG8 antibody is available, the authors can detect the ATG8 expression level in this mutant.

First revision

Author response to reviewers' comments

Reviewer 1:

SUMMARY OF THE ADVANCE MADE IN THIS PAPER AND ITS POTENTIAL SIGNIFICANCE TO THE FIELD
Although the autophagy machinery is highly conserved between major taxonomic groups, ATG8, a central autophagy gene, has undergone expansion uniquely in plants. This interesting short report provides convincing evidence to support the hypothesis that expansion of ATG8 underpins functional specialisation.

The authors successfully created a nonuple *atg8* mutant through two rounds of CRISPR-Cas9 genome editing and then tested complementation of different mutant phenotypes by representatives of two distinct ATG8 clades. The study employs well established assays to demonstrate that both ATG8A and ATG8H mediate autophagic recycling during carbon starvation but only 8A recovers sensitivity of the *atg8* nonuple mutant to nitrogen starvation. Extending this observation, proximity ligation suggests that the two ATG8 isoforms may interact with distinct groups of proteins. These interactions were not validated by an orthogonal technique, which would be required for a larger, more in-depth study but not in my opinion for a short report. The proxitome data are in agreement with a study of potato ATG8 from some of the same authors (Zess et al., 2019). Whilst supportive, this detracts to some extent from the novelty of the current report. However, the manuscript answers a key question in the field and provides a valuable tool for plant autophagy research.

SUGGESTIONS TO AUTHORS

TEM was used to demonstrate lack of mitophagy in the *atg8* nonuple mutant, with representative images presented. However, I could not find details of how many sections were examined and how prevalent the mitophagosomes are in wild type. It would be helpful to provide details of the sampling and any quantification strategy to give confidence in the result.

For each sample (WT and *atg8*), more than 30 sections from three individual roots were examined. As we were focused on checking whether *atg8* has autophagosome or not, not all the autophagosomes or mitophagosomes in WT were captured (only some representative images were captured). This information is now included in the Materials and Methods section.

The effect of ATG8H is not as marked as that for ATG8A in Fig. 4C; perhaps this could be mentioned in the text?

The purpose of these assays is to assess the ability of single ATG8 isoforms to recover, even

partially, the wild-type phenotype in the $\Delta atg8$ mutant. Our tests reveal that complementation with either ATG8A or ATG8H allows partial recovery of NBR1 flux under C starvation. In the attempt to evaluate differences in the capacity of the two isoforms to recover the wild-type phenotype, NBR1 endogenous levels were evaluated side by side in the ATG8A and the ATG8H complemented lines. We can observe that in control conditions (+C) NBR1 appears to accumulate more in the ATG8H complemented line, however under C starvation both complemented lines exhibit comparable NBR1 levels. (Not included in the manuscript, just one replicate)

Please provide a reference for mitophagy in HeLa cells lacking ATG8 (bottom of page 6).

Apologies for this. We have added the reference.

Reviewer 2: SUMMARY OF THE ADVANCE MADE IN THIS PAPER AND ITS POTENTIAL SIGNIFICANCE TO THE FIELD

The manuscript entitled "Nonuple atg8 mutant provides genetic evidence for functional specialization of ATG8 isoforms in Arabidopsis thaliana" describes the functional diversification of the nine ATG8 homologs in Arabidopsis by generating a complete knock-out mutant of ATG8. Their results show that Arabidopsis ATG8 homologs are differentially expressed and could have functional diversification. The generated nonuple atg8 KO mutant will be an important tool for the whole community and the findings described in this manuscript are of interest to a wide range of plant scientists.

SUGGESTIONS TO AUTHORS

The manuscript is very well written and I have only a few comments, mainly regarding the nonuple atg8 mutant. For its use in future research, it would be important that some aspects of the mutant are described in detail.

1. Figure 1A: The paper by Thompson et al (DOI: 10.1104/pp.105.060673) also investigated the differential induction of ATG8 homologs. In addition, a previous review by Woltering et al (DOI: 10.3389/fpls.2020.579875) describes the differences in mRNA and protein abundance of ATG8 homologs based on expression and proteomics data and Chung et al (doi: 10.1104/pp.108.126714) investigates the ATG8 homologs in maize. These papers should be cited for discussing the results shown here.

The mentioned papers are now cited and discussed in relation to the results presented.

2. Figure 1B: I may have missed the description, but it was not clear to me under which promoter the GFP/mCherry ATG8 homologs were expressed. The methods only describe pGGSUN_HTR5::GFP-ATG8A and pGGSUN_HTR5::GFP-ATG8H. If non-endogenous promoters are used, a colocalization assay may not reflect physiological situations due to different expression patterns. The authors should describe the experimental set-up and may wish to comment on this.

The lines used for the colocalization assays have both GFP- and mCherry-ATG8X fusions driven by the UBQ10 promoter. The information is now added in the Plant material and cloning paragraph. The constitutive promoter UBQ10 allowed uniform labelling of autophagosomes by all ATG8 isoforms and yet, different colocalization ratios between mCherry-ATG8E and different GFP-ATG8

isoforms were observed, which emphasizes that ATG8 isoforms have distinct functions. Nevertheless, in Figure 1A we already showed that all ATG8 isoforms are expressed at the epidermal cells of the transition zone in the root, so our colocalization experiments are physiologically relevant. However, we agree with the reviewer that expression polymorphisms will also contribute to functional specialization, which we have addressed with the transcriptional reporters.

3. Fig. 2A and in general to the nonuple atg8 mutant: Although the data indicate that the generated nonuple mutant is a complete knock-out mutant, some additional information should be provided. It would be helpful if the authors described the translational consequences of the indel events for each homolog. For ATG8B, the 6 bp deletion that does not cause a frameshift and generate a premature stop codon. Here, it is not clear whether a 2 amino acid deletion affects ATG8B function. If ATG8B is still intact, the atg8 mutant is an octuple instead of a nonuple mutant. This doesn't diminish the importance of this genetic material or the impact of this study, but it should be clarified for future use. Can the authors map the position of the deletion in the protein structure and speculate? qPCR should be performed to investigate whether ATG8B transcripts are still present in the mutant as non-functional mRNA would decay.

The 6 bp deletion falls at the exon-intron junction on exon two of the ATG8B gene and the mutation results in a premature truncation of the protein, as illustrated in Supplementary table 1.

4. Figure 3C and page 4, line 56 to page 5, line 9/page7 lines 1-2 "... we did not observe any mitophagosomes in Δ atg8 mutant (Fig. 3C). However, we could still detect double-membrane structures that resemble autophagosomes in the Δ atg8 mutant." These sentences are a bit too speculative. It has not been investigated in plants whether and to what extent phagophores or even autophagosomes (without ATG8) can be formed. Although more specialised in function, ATG8-independent autophagy could still occur. The localisation of non-ATG8 autophagosomal markers in the atg8 mutant and provacuole markers should be tested to clarify this. If this is not possible, the sentence should at least be rephrased so that the formation of autophagosomes is not categorically excluded.

We have rephrased this paragraph.

5. Figure 4A/B: The expression levels of ATG8A and ATG8H should be examined. Otherwise, it cannot be excluded that the difference in complementation of the Δ atg8 mutant simply reflects the level of expression rather than the differences in the two homologs.

A supplemental figure was added (Fig. S2C) providing qPCR results for the two complemented lines, showing comparable levels of expression of the fusion proteins GFP-ATG8A and GFP- ATG8H.

6. Figure 4D: The presentation makes it difficult to see the differences between carbon and nitrogen starvation. The authors may consider presenting the quantification separately, if possible, by including quantification of all replicates.

The quantifications of the relative protein abundance are now presented separately in boxplots.

7. Figure 4E. The Turbo-ID data are highly interesting but would need further validation to be truly informative. The authors may even want to consider removing this data from the current manuscript as the main conclusions seems mostly well supported by the rest of the data. If they decide to include the data, it is understandable, but the experiment should be described a little more in detail.

Are the expression levels of the Turbo-ID constructs for ATG8H and ATG8A similar? Can at least some of the "differential" interactor candidates be verified?

The protein abundance of the Turbo-ID fusions for ATG8A and ATG8H is comparable, the western blot is now presented in Supplementary figure 2D. The validation of the TurboID candidates is beyond the scope of this short report. We and others in the community are working hard to characterize these new interactors in depth.

Reviewer 3: SUMMARY OF THE ADVANCE MADE IN THIS PAPER AND ITS POTENTIAL SIGNIFICANCE TO THE FIELD

In this study, the authors have generated transcriptional and translational ATG8 reporters in *Arabidopsis thaliana* to reveal their expression and function diversity, which has not been addressed for a long time.

SUGGESTIONS TO AUTHORS

1. Using nonuple $\Delta atg8$ mutant, the authors showed that it is hypersensitivity to carbon and nitrogen starvation. However, in a recent publication (<https://doi.org/10.1038/s41467-024-55754-1>), this group has used the same mutant to demonstrate its sensitivity to nitrogen starvation and the functional diversity between ATG8f and ATG8i, which corresponding to ATG8A and ATG8H used in this study. They have claimed that *Arabidopsis* ATG8 isoforms differ in their capacity to carry out autophagic flux. The authors need to cite this study and remove the overlapping result or to supplementary data.

The indicated paper is now cited in the manuscript and the results are discussed in relation to their finding of ATG8I inability to respond to nitrogen starvation, likewise to ATG8H that we present. Since we used different ATG8 isoforms for complementation, the results are not overlapping, rather complementary.

2. Figure titles need further editing to describe concisely. For example, 'Supplementary Figure 2. *Arabidopsis* ATG8 complementation lines have normal autophagosome structure and autophagic flux under control and nutrient-deficient conditions.' If they are normal, the overall conclusion for functional specification is incorrect.

What we claim is that different ATG8 isoforms are functionally specialized, which means they may respond to different stimuli or be able to interact with distinct protein partners to carry out distinct cellular recycling mechanisms. The observation that both complemented lines are able to form autophagosomes does not contradict the hypothesis.

"The $\Delta atg8$ mutant is not able to perform mitophagy and pexophagy." In comparison with *atg5* mutant, it seemed that *atg8* mutant contains less IDH, VDAC and catalase after treatment. Better to quantify the immunoblot result.

The relative protein abundance of IDH, VDAC and catalase was calculated and is now presented in boxplots in Figure 3. The levels of IDH and VDAC are comparable in the *atg8* and *atg5* mutants. As for catalase, *atg8* might accumulate lower levels of catalase compared to the *atg5* mutant, but the phenotype compared to the wild-type still reveals an impairment of pexophagy.

3. To my understanding, functional specialization might be related to autophagosome formation, which has been reported for human ATG8 orthologs, from expansion, maturation to completion. Certainly, it seemed that the proxitomes for ATG8A and ATG8I are different, but there is no verification for any candidates, at least one example can be tested. Though it seemed that not all ATG8 isoforms overlaps with ATG8e, but whether they all have similar expression level as ATG8e?

We have expressed the ATG8 isoforms with the same promoter (UBQ10) to ensure similar expression levels. Characterization of candidates is beyond the scope of this short report.

4. As there are still some puncta colocalized with ATG8e, real time imaging to show whether they are sequentially recruited to the autophagosome will be useful to provide more information for their functional specialization during autophagosome formation. Otherwise, it is possible that the autophagosome has already been decorated enough ATG8 proteins.

As we mentioned above, further characterization of ATG8 functional specialization in terms of autophagosome formation or maturation is beyond the scope of this short report. We and other colleagues in the community are investigating these hypotheses in depth and will present our findings in follow up manuscripts.

5. For the complementation lines, both ATG8A and ATG8i were driven by the strong HTR5 promoter. As shown in Figure 1A, different isoforms showed various expression patterns in plant tissues. How to explain on the medium with carbon, GFP-ATG8A and GFP-ATG8i both were already degraded?

We used constitutive promoter to avoid differences in expression, since our goal was to understand if both ATG8 isoforms are present in similar levels, could they complement nutrient starvation sensitivity. Both of them being degraded simply means they can still localize to autophagosomes and undergo basal autophagic flux, which serves as a positive control for that experiment. So, we don't see any issues with this experiment.

6. For the *atg8a-i* mutant, as ATG8 antibody is available, the authors can detect the ATG8 expression level in this mutant.

A western blot comparing ATG8 proteins expression in the wild-type and Δ atg8 mutant is now presented in Supplementary figure 1.

Second decision letter

MS ID#: jcs.263803R1

MS TITLE: Nonuple *atg8* mutant provides genetic evidence for functional specialization of ATG8 isoforms in *Arabidopsis thaliana*

AUTHORS: Alessia Del Chiaro; Nenad Grujic; Jierui Zhao; Ranjith K. Papareddy; Peng Gao; Juncai Ma; Christian Loeffke; Anuradha Bhattacharya; Ramona Gruetzner; Pierre Bourguet; Frederic Berger; Byung-Ho Kang; Sylvestre Marillonnet; Yasin Dagdas

ARTICLE TYPE: Short Report

Dear Dr Dagdas,

We have now reached a decision on the above manuscript.

To see the reviewers' reports and a copy of this decision letter, please go to:

As you will see, the reviewers (and myself) appreciate the thorough revision of the original manuscript. The reviewers raised some minor points that I hope you will be willing to address. I would like to point out that at this stage, I expect only light edits to the manuscript and no further experiments. I am looking forward to being able to accept your paper pending these changes.

Second revision

Author response to reviewers' comments

Reviewer 2: The authors clarified all the points. Supplementary Table 1 should be called together with Figure 2A and Fig S1A in page 4 line 84. Apart from that I do not have any further comments.

The reference to Supplementary Table 1 has been added in the indicated paragraph.

Reviewer 3: SUMMARY OF THE ADVANCE MADE IN THIS PAPER AND ITS POTENTIAL SIGNIFICANCE TO THE FIELD

In this study, the authors have generated transcriptional and translational ATG8 reporters in *Arabidopsis thaliana* to reveal their expression and function diversity, which has not been addressed for a long time.

SUGGESTIONS TO AUTHORS

Fig 3. Based on the protein levels the authors claims that mitophagy and pexophagy are not performed in $\Delta atg8$ mutant, but only EM showing mitophagosome or mitochondria was presented. I would suggest to remove the conclusion for pexophagy as no other data support it. Instead, an example for a Golgi-like structure in the $\Delta atg8$ mutant was shown. In that case, why only selective autophagy is specifically affected in mitophagy and pexophagy? or include Golgi?

We modified that sentence to downgrade our conclusion: Although further characterization with multiple pexophagy and mitophagy are required, these results suggest $\Delta atg8$ mutant is defective in mitophagy and pexophagy.

The authors claims that "ATG8 isoforms are differentially expressed across tissues and form distinct autophagosomes", but when referring "functional specification", it is only related to stress treatment or nutrient starvation. To me, as they form distinct autophagosomes for different cargoes for recycling and therefore leads to "functional specification".

"We have expressed the ATG8 isoforms with the same promoter (UBQ10) to ensure similar expression levels." I don't agree with that UBQ10 promoter can ensure similar expression levels for different transgenic lines. The authors can measure by immunoblotting.

The reviewer would be happy to know that when selecting these lines, we did the western blots they are suggesting and chose lines that express at similar levels.

Minor comments

Fonts for all quantifications for the mutants are not in italic.

Labels for the mutants have been changed to italic.

Third decision letter

MS ID#: jcs.263803R2

MS Title: Nonuple atg8 mutant provides genetic evidence for functional specialization of ATG8 isoforms in *Arabidopsis thaliana*

Authors: Alessia Del Chiaro; Nenad Grujic; Jierui Zhao; Ranjith K. Papareddy; Peng Gao; Juncai Ma; Christian Loeffke; Anuradha Bhattacharya; Ramona Gruetzner; Pierre Bourguet; Frederic Berger; Byung-Ho Kang; Sylvestre Marillonnet; Yasin Dagdas

Article Type: Short Report

Dear Dr Dagdas,

I am happy to tell you that your manuscript has been accepted for publication in Journal of Cell Science, pending standard publication integrity checks.